# Ketoprofen Lysine Salt Versus Corticosteroids in Early Outpatient Management of Mild and Moderate COVID-19: A Retrospective Study

**DOI:** 10.3390/pharmacy13030065

**Published:** 2025-05-01

**Authors:** Domenica Francesca Mariniello, Raffaella Pagliaro, Vito D’Agnano, Angela Schiattarella, Fabio Perrotta, Andrea Bianco

**Affiliations:** Department of Translational Medical Sciences, University of Campania ‘L. Vanvitelli’, 80131 Naples, Italy; nikamariniello93@gmail.com (D.F.M.); vito.dagnano@studenti.unicampania.it (V.D.); angela.schiattarella@studenti.unicampania.it (A.S.); andrea.bianco@unicampania.it (A.B.)

**Keywords:** ketoprofen lysine salt, NSAIDs, COVID-19, SARS-CoV-2

## Abstract

*Background:* Accelerating recovery and preventing the progression to more severe outcomes for patients with coronavirus disease 2019 (COVID-19) is of paramount importance. Non-steroidal anti-inflammatory agents (NSAIDs) have been widely adopted in the international recommendations for non-severe COVID-19 management. Among NSAIDs, evidence about the efficacy of ketoprofen lysin salt (KLS) in the treatment of non-severe COVID-19 has not been reported. *Methods:* This retrospective study compared the outcomes of 120 patients with mild to moderate COVID-19 treated at home with KLS between March 2021 and May 2023 compared with the outcomes of 165 patients who received corticosteroids. The outcomes included hospitalization, the need for oxygen supplementation, clinical recovery from acute COVID-19, and time to negative swabs. *Results:* Symptoms persisted in a lower percentage of patients in the KLS group compared to the corticosteroids group (*p* < 0.0001) and for a shorter period (*p* = 0.046). We found 6 patients (5%) in the KLS group were hospitalized compared to 45 (27%) in the corticosteroids group (*p* < 0.001). A higher percentage of patients in the corticosteroids group require oxygen administration (*p* < 0.001). In addition, patients taking corticosteroids showed a longer viral shedding period compared to those taking KLS (*p* = 0.004). A final multivariate analysis suggests that KLS might reduce hospitalization risk, the need for oxygen supplementation, and the persistence of post-COVID-19 symptoms when compared to an oral corticosteroid after adjusting for significant co-variables. *Conclusions:* KLS might have a positive effect on clinical recovery in non-severe COVID-19 patients. A comparison with other NSAIDs in terms of difference in efficacy and safety should be investigated in further trials.

## 1. Introduction

The severe acute respiratory syndrome coronavirus 2 (SARS-CoV-2), responsible for the novel coronavirus disease 2019 (COVID-19), has led to a global public health emergency [1].

Although infection with SARS-CoV-2 can be asymptomatic, most patients commonly present with a wide spectrum of symptoms such as fever, sore throat, dry cough, malaise, muscle pain, shortness of breath, gastrointestinal symptoms, and loss of taste or smell. Disease in some patients can progress to severe/critical conditions, such as interstitial pneumonia and acute respiratory distress syndrome (ARDS), and death [2,3,4]. Many patients with severe COVID-19 have an excessive inflammatory response caused by an uncontrolled release of proinflammatory cytokines, defined as a cytokine storm [5,6]. The elevated levels of inflammatory markers, including C-reactive protein, ferritin, interleukin (IL)-1B, and IL-6, represent the signature of a severe COVID-19 phenotype [7]. Targeting inflammation with anti-inflammatory drugs early in the course of symptomatic disease may be a suitable strategy for managing COVID-19 disease severity. While the management of hospitalized patients with severe COVID-19 is well-established, there is little knowledge about therapeutic strategies in early COVID-19 to prevent progression.

Non-steroidal anti-inflammatory drugs (NSAIDs) are routinely prescribed for their analgesic, anti-inflammatory, or antipyretic effects. NSAIDs might be beneficial for the early management of COVID-19 in the outpatient setting to manage symptoms and prevent the progression to a more severe disease although the use of NSAIDs in COVID-19 was debated at the start of the COVID-19 pandemic. The initial concerns originated from anecdotal reports that suggested that NSAIDs (particularly ibuprofen) could increase the susceptibility to infection and adversely impact the prognosis of COVID-19 patients through the upregulation of angiotensin-converting enzyme 2 (ACE2) receptor, the entry receptor for SARS-CoV-2 [8,9,10,11]. Later, several studies concluded that NSAID use was not significantly associated with a higher risk of SARS-CoV-2 infection or with worse outcomes [12,13,14,15]. Some NSAIDs, in addition to their anti-inflammatory and analgesic proprieties, have been reported, often at high doses, to have antiviral activity in vitro, which has been suggested to contribute to their efficacy in the treatment of COVID-19 [16]. Among these molecules, it is well-known that ketoprofen is more effective in reducing inflammation than other common NSAIDs like ibuprofen, phenylbutazone, and aspirin [17,18]. The salification of ketoprofen with L-lysine has led to the synthesis of ketoprofen lysine salt (KLS), a non-selective inhibitor of cyclo-oxygenase (COX) and lipoxygenase pathways [19]. This formulation improves solubility, absorption, and the onset of therapeutic effects, and provides better gastrointestinal tolerability compared to ketoprofen [20,21,22]. Moreover, KLS can penetrate extensively into the upper airways [23]. However, real-world data about effectiveness of ketoprofen lysine salt (KLS) is still lacking in the current literature [18,24].

Regarding corticosteroid use, based on the data of the RECOVERY trial, dexamethasone has been shown to reduce COVID-19 28-day mortality only among patients who were receiving respiratory support; more controversial is the use of glucocorticoids in patients who were not receiving respiratory support [25]. Very little evidence is available regarding the effectiveness of the early administration of glucocorticoids in COVID-19 patients with mild disease in the outpatient setting. Moreover, the use of corticosteroids in COVID-19 patients seems to delay the time of viral clearance and to increase the potential risk of secondary infections [25,26,27].

Since an adequate host immune response is required in viral infections, the use of NSAIDs or glucocorticoids during the early viral response to SARS-CoV-2 could worsen the outcome of the disease, so it was unclear whether a mitigated immune response would be beneficial, harmful, or neutral in the setting of COVID-19 [28].

This study aimed to compare the disease course and the outcomes of patients with mild to moderate COVID-19 treated with KLS or corticosteroids during the early phase of infection.

## 2. Methods

We conducted a single-center retrospective study involving symptomatic patients with a documented positive rhino-pharyngeal swab for SARS-CoV-2 RNA admitted to outpatient service for COVID-19 at U.O.C. Pneumologia “L. Vanvitelli” A.O. dei Colli—Ospedale Monaldi, Naples, Italy between March 2021 to May 2023. Adult individuals (age ≥ 18 years) with mild to moderate COVID-19 were included in the study and they were divided in two subgroups according to anti-inflammatory drugs used in the early phase (symptom onset) of the infectious disease: KLS and corticosteroids groups. Disease severity was determined according to the World Health Organization (WHO) as mild, moderate, severe, or critical [29]; in particular, mild illness has been defined as individuals who have any of the various signs and symptoms of COVID-19 (e.g., fever, cough, sore throat, malaise, headache, muscle pain, nausea, vomiting, diarrhea, and loss of taste and smell) but do not have shortness of breath, dyspnea, or abnormal chest imaging, while moderate illness is defined as individuals who show evidence of lower respiratory disease during clinical assessment or imaging and who have an oxygen saturation measured by pulse oximetry (SpO2) ≥ 94% on room air at sea level. Subjects who required immediate hospital admission because of severe COVID-19 symptoms at onset were excluded. The dose of KLS was 80 mg twice day; corticosteroids were administered in accordance with current guidelines, mainly in mild to moderate COVID-19 patients presenting with significant respiratory symptoms. KLS was mainly prescribed for patients experiencing myalgia or in cases where corticosteroid use was contraindicated. No standardized compliance measurement was performed due to outpatient setting.

If patient’s symptoms improved over time, the need for ongoing therapy, including KLS administration, was re-evaluated and treatment was either tapered down or discontinued [30,31]. At outpatient visit, the following data of the entire study group were collected from medical records on an electronic database: (a) demographic characteristics (age and sex); (b) smoking status (actual and former or never smokers) and pack years; (c) comorbidities—in particular, the Charlson comorbidity index (CCI) was calculated by summing the assigned weighted score of 19 comorbid conditions: higher scores indicated a more severe condition and, consequently, a worse ten-year survival [32]; (d) SARS-CoV-2 vaccination status: fully vaccinated or unvaccinated/partially vaccinated; (e) COVID-19 symptoms in the acute phase of the infection: fever, sore throat, dyspnea, muscle pain, anosmia or ageusia, diarrhea, and fatigue; (f) need for oxygen supplementation and/or hospitalization (intensive care unit (ICU) admission, mechanical ventilation); (g) the median time to resolution of major symptoms (complete remission); (h) time of persistence of symptoms after recovery from acute COVID-19; (i) long COVID-19 symptoms (long COVID was defined as having at least one of 25 WHO-listed symptoms between 90 and 365 days after the date of a PCR-positive test or clinical diagnosis of COVID-19, with no history of that symptom 180 days before SARS-CoV-2 infection) [33,34]; (j) serious (SAEs) and non-serious adverse events (AEs) related to the administered treatments; and (k) chronic concomitant medications due to pre-existing comorbidities. The main outcomes were hospitalization, oxygen supplementation, clinical recovery from acute COVID-19, and time to negative swabs. The study was approved by the local ethics committee and was in accordance with the 1976 Declaration of Helsinki and its later amendments. Informed consent was obtained from all participants in outpatient department before the enrolment. The ethics committee was that of the Azienda Ospedaliera dei Colli, Napoli, Italy with number 16223/2020 and later amendment.

## 3. Statistical Analysis

Continuous variables are expressed as the means with standard deviations if they conformed to a normal distribution, or medians with interquartile ranges (IQRs) if they did not. The differences between groups for continuous variables were compared using the unpaired Student’s *t*-test (normal distribution) or Wilcoxon–Mann–Whitney tests (non-normal distribution). Categorical variables, reported as counts and percentages, were compared between groups using the χ^2^ test. For ordinal or non-normally distributed continuous variables, Spearman’s rank correlation test was used.

To assess the influence of KLS on key clinical outcomes (hospitalization risk, need for supplemental oxygen, and persistence of post-COVID-19 symptoms), a multivariable binomial logistic regression analysis was performed using a backward elimination procedure based on *p*-values. All analyses were conducted using STATA software, version 16 (StataCorp. 2019. College Station, TX, USA). A *p*-value < 0.05 was considered statistically significant.

## 4. Results

A total of 285 patients were included in the study, 120 patients in the KLS group and 165 patients in the corticosteroids group. None of the patients underwent a treatment switch from KLS to corticosteroids, or vice versa. The baseline patient characteristics were generally comparable among the two treatment groups, as shown in the table (Table 1).

### 4.1. Patient Characteristics

By comparing the clinical characteristics of patients in the KLS group with those of patients in the corticosteroids group, there was no significant difference in age (67.8 vs. 66.0, *p* = 0.402) and both groups had a slight prevalence of males. While the distribution of concomitant diseases showed a slight increased prevalence of patients with coronary artery disease (CAD) or chronic heart failure in the group treated with corticosteroids, patients with type II diabetes were less likely to receive corticosteroids. The overall burden of comorbidities between the two groups was similar as the Charlson comorbidity index was not significantly different (3.0 vs. 2.0, *p* = 0.177). The percentage of current or former smokers was significantly higher in the KLS group in comparison with the corticosteroids group (78% vs. 60%, *p* = 0.025). There was no significant difference in SARS-CoV-2 vaccine status (*p* = 0.535). The most common symptoms at the onset of disease in both groups were sore throat, fever, and dyspnea.

### 4.2. Main Clinical Outcomes

A higher percentage of patients in the corticosteroids group required hospitalization (27% vs. 5%, *p* < 0.001; Figure 1a) and the need for oxygen supplementation (42% vs. 10%, *p* < 0.001; Figure 1b) in comparison with the KLS group. Patients taking corticosteroids showed a longer time to a negative swab compared to those taking KLS (23.8 ± 11.0 vs. 17.7 ± 9.40 days, *p* = 0.004). Symptoms persisted in a lower number of patients (33% vs. 66%, *p* <  0.001; Figure 1c) and for a shorter period (133 ± 123 vs. 194 ± 165, *p* = 0.046) in the KLS cohort compared with the corticosteroid cohort. The relative risk for developing the main clinical outcomes in the study population cohort is reported in Table 2. In the final adjusted logistic multivariate analysis (Table 3), the use of KLS was associated in fewer hospitalizations, a reduced need for oxygen supplementation, and a reduced prevalence of prolonged long COVID-19 symptoms (*p* < 0.001). The collinearity statistical analysis is reported in Appendix A.

## 5. Discussion

To the best of our knowledge, this is the first study to explore the role of KLS for the early outpatient management of COVID-19 symptoms. In particular, we explore whether the use of KLS influences the progression to more severe disease. We found that the early administration of KLS in COVID-19 may prevent the progression to a more severe disease and long-term complications in comparison with corticosteroid treatment. NSAIDs inhibit the activity of cyclo-oxygenase enzymes, namely, COX-1 and COX-2 [35]. The inhibition of COX-1 and COX-2 suppresses the formation of prostanoids, including prostaglandin E2 (PGE2), D2 (PGD2), and F2α (PGF2α), thromboxane A2 (TxA2), and prostacyclin (PGI2), key inflammatory mediators. Prostanoids also have other important biological effects including vascular tone, platelet function, kidney function, and gastrointestinal protection [36]. Elevated levels of PGE2, PGF2α, and TxA2 have been found in biological samples from inpatients with COVID-19 compared to healthy controls [37,38,39,40]. SARS-CoV-2 itself seems to upregulate the expression of genes encoding for COX-1 and COX-2, resulting in a significant accumulation of prostaglandins [41,42,43]. A study longitudinally performed a peripheral blood RNA sequencing analysis of the same patient’s samples at different time points between symptom onset and recovery from COVID-19. The authors found an increased expression of COX-2 at the early onset of disease and at the clinically most severe stage [41].

Some NSAIDs have been reported, often at high doses, to have antiviral activity in vitro, which has been supposed to contribute to their efficacy in the treatment of COVID-19 [44]. For example, Xu et al. found that indomethacin has potent antiviral activity against SARS-CoV-2 in vitro and in vivo, and that recovery occurred significantly sooner with a combination of symptomatic treatment and indomethacin in infected dogs [45]. Moreover, Sisakht et al. demonstrated that NSAIDs downregulated the expression of prostaglandin E synthase, and it is well-known that PGE2 increases the viral pathogenicity by contributing to hyperinflammatory activity, as well as affecting the viral transcription, translation, and/or replication. In addition, the authors showed that NSAIDs acted as inhibitors of the 3CLpro, the main protease of the SARS- CoV-2 involved in the proteolytic maturation of the virus [46].

The hypothesis that NSAIDs may be beneficial in COVID-19 if used regularly during the first days from the onset of symptoms has recently been proposed by Perico et al., who present a home treatment protocol that includes the use of NSAIDs, in particular, relatively selective COX-2 inhibitors [47]. The authors suggest to start NSAIDs at the onset of symptoms (fever, cough, sore throat, or headache) for three to four days and continue if symptoms persist. Should symptoms persist after eight to ten days of NSAIDs or oxygen saturation significantly decline, oral dexamethasone should be commenced and NSAIDs discontinued.

Some observational studies have looked at NSAIDs, especially relatively selective COX-2 inhibitors, for the early home treatment of COVID-19 patients. COX-2 plays a key role in the pathogenesis of viral infections; in a model of influenza A infection, compared with wild-type or COX-1-deficient mice, COX-2-deficient mice had better survival rates and lower levels of proinflammatory cytokines despite having higher levels of virus [48].

A retrospective observational study compared the outcomes of two matched cohorts of patients with mild-to-moderate COVID-19 treated at home by their family physicians with relatively selective COX-2 inhibitors (nimesulide or celecoxib) vs. those receiving other therapeutic regimens, such as paracetamol. In the cohort of patients treated with selective COX-2 inhibitors, symptoms persisted less frequently and for a shorter period, fewer patients were hospitalized, and the cumulative costs of hospitalization were reduced by >90% [49]. Although NSAIDs which selectively inhibit COX-2 had less gastrointestinal toxicity, they were associated with an increased risk of cardiovascular events [50,51]. Moreover, Nimesulide has been associated with a risk of hepatotoxicity [52].

Ketoprofen is a highly potent and safe NSAID of the propionic acid derivative group; it is a non-selective NSAID but expresses high activity on COX-2 [53,54]. Ketoprofen also inhibits the bradykinin’ production and the lipoxygenase pathway of the arachidonic acid cascade, reducing the release of non-cyclized mono-hydroxy acids (HETE) and leukotrienes [55].

The salification of ketoprofen with the lysine amino acid allows for a higher solubility that facilitates a more rapid and complete absorption of the drug and determines a better gastrointestinal tolerability due to the reduced effective daily dosage [24,56,57]. In addition, KLS has both peripheral and central activity, inhibiting both nitric oxide (NO) and COX synthase in the brain [58,59]. Moreover, KLS is able to penetrate extensively into the upper airways, improving sore throat symptoms, and in the adipose tissue; the latter is particularly attractive because recent data demonstrated that subjects with severe COVID-19 have higher local visceral adipose tissue inflammation [23,60,61,62]. Another distinctive pharmacological effect of KLS is its antiplatelet effect with a potency superimposable to that of aspirin [63]. This activity is particularly important in the management of COVID-19 disease as, during the cytokine storm, the vessels’ endothelium is activated, reducing prostacyclin and NO production, important antiaggregant mediators, causing diffuse coagulopathy [64,65,66,67].

According to international guidelines, glucocorticoids are the mainstay of treatment for severe and critical COVID-19 patients who are mechanically ventilated or who require supplemental oxygen [68]. The role of corticosteroids is controversial in outpatients in the early phase of COVID-19 infection because, in this phase, they may suppress host antiviral responses [69]. National Institutes of Health (NIH) guidelines recommend against the use of systemic corticosteroids in not-hospitalized patients with mild to moderate COVID-19. Similarly to our results, a systematic review showed that corticosteroid treatment in mild or moderate COVID-19 patients is associated with a more extended hospitalization and more days of viral shedding [70]. A retrospective study investigated the effects of methylprednisolone in patients with moderate COVID-19, showing that this treatment significantly prolonged the hospital stay and time from admission to viral clearance [71].

The efficacy of inhaled corticosteroids in patients with early COVID-19 remains to be clarified because persistent inflammation in the upper airways seems to be potentially associated with the progression to severe COVID-19 [72]. The STOIC study showed that inhaled budesonide reduced the need of urgent medical care and persistent long-term symptoms in COVID-19 (long COVID) compared to the usual care [73]. These results have also been confirmed by the PRINCIPLE study among people at a high risk of complications, although statistical significance was not achieved for the endpoint of hospitalization [74]. Conversely, the CONTAIN study revealed that the combination of inhaled and intranasal ciclesonide did not improve respiratory symptoms, nor did it reduce the incidence of hospitalization in outpatients with early-stage COVID-19 compared to placebo [75].

Since the majority of patients with COVID-19 experience a mild or moderate illness and do not require hospital admission, the optimal management of outpatients by primary care physicians is of paramount importance. Therefore, the appropriate time and dosage of using nonsteroidal anti-inflammatory drugs in COVID-19 patients should be carefully evaluated.

### Study Limitations

We acknowledge that our study presents several limitations. Firstly, the non-randomized design and the retrospective nature of the statistical analyses did not allow for robust information in order to assess the difference between groups and a prospective randomized control study should be conducted to confirm our findings. Furthermore, based on the methodology, we did not assess baseline hematological blood parameters, such as markers of inflammation; therefore, we cannot exclude that the groups might be different in inflammatory parameters whose potentially influence the prognosis of COVID-19 patients. The odds ratios interpretation should be considered with caution as we cannot exclude, for some variables with high standard errors, a separation issue. Therefore, future prospective research should verify these data and also assess the treatment dosage and compliance in rigorous blinded clinical trials.

## Figures and Tables

**Figure 1 pharmacy-13-00065-f001:**
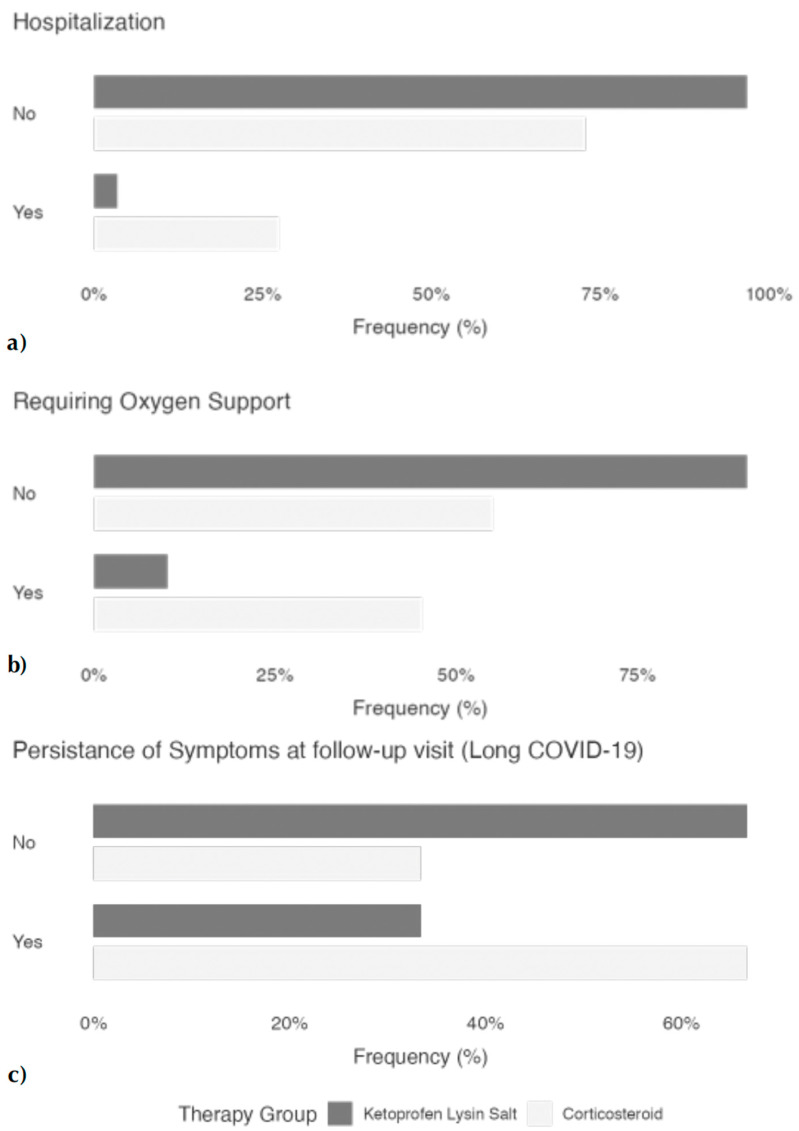
Main COVID-19 outcomes in non-severe patients. (**a**) Patients treated with KLS compared with patients treated with corticosteroids were less likely to be hospitalized (*p* < 0.001); (**b**) Patients treated with KLS compared with patients treated with corticosteroids were less likely to require oxygen support (*p* < 0.001); (**c**) patients treated with KLS compared with patients treated with corticosteroids were less likely to have longCOVID-19 symptoms (*p* < 0.001).

**Table 1 pharmacy-13-00065-t001:** Baseline and clinical characteristics of population study.

	*KLS Group (n = 120)*	*Corticosteroids Group (n = 165)*	*p*
** *Males, n (%)* **	65 (54)	85 (51.5)	0.658
** *Age, years (mean ± SD)* **	67.8 [49.8–71.3]	66.0 [56.0–72.0]	0.402
** *Pack years* *(mean ± SD)* **	30.0 [20–52]	25.0 [22.5–30]	0.160
** *Smoker Status* **			0.025
*Current/Former, n (%)*	85/120 (70.8)	100/165 (60.6)	
*Never, n (%)*	35/120 (29.2)	65/165 (39.4)	
*Unknown*	5/120 (4.2)	0/165 (0)	
** *Comorbidities* **			
*COPD*	22/120 (18.3)	26/165 (15.8)	0.57
*Systemic Hypertension*	51/120 (42.5)	80/165 (48.5)	0.32
*Coronary Artery Disease or Chronic heath failure*	11/120 (9.2)	30/165 (18.2)	0.03
*CKD*	2/120 (1.7)	8/165 (4.8)	0.15
*Malignant Neoplasms*	5/120 (4.2)	2/165 (1.2)	0.11
*Type II Diabetes*	36/120 (30.0)	25/165 (15.1)	<0.01
*Hypercolesterolemia*	43/120 (35.8)	45/165 (27.3)	0.12
*Stroke or Dementia*	4/120 (3.3)	3/165 (1.8)	0.41
*Hematological Disorders*	15/120 (12.5)	10/165 (6.1)	0.06
** *Charlson Comorbidity Index* *(mean ± SD)* **	3.0 [2–5]	2.0 [1–4]	0.177
**SARS-CoV-2 Vaccine Status, *n (%)***			0.897
*Fully Vaccinated*	60/120 (50)	85/165 (51.5)	
*Unvaccinated/Partially Vaccinated*	60/120 (50)	80/165 (48.5)	
** *Early COVID-19 Symptoms, n (%)* **			
*Fever*	65/120 (54.2)	90/165 (54.5)	0.95
*Sore throat*	56/120 (46.7)	85/165 (51.5)	0.42
*Dyspnea/Chest Tightness*	48/120 (40.0)	113/165 (68.5)	**<0.001**
*Muscle or body aches*	69/120 (57.5)	96/165 (58.2)	0.91
*Loss of smell/Taste*	22/120 (18.3)	38/165 (23.0)	0.34
*Diarrhea*	13/120 (10.8)	18/165 (10.9)	0.98
*Fatigue*	28/120 (23.3)	44/165 (26.6)	0.52
** *Patients with long COVID Symptoms (Yes), n (%)* **	40/120 (33)	110/165 (67)	**<0.001**
** *Long COVID Symtpoms, n (%)* **			
*Dyspnea/Chest Tightness*	35/40 (87.5)	95/110 (86.4)	0.848
*Cough*	20/40 (50.0)	58/110 (52.7)	0.776
*Tachycardia*	5/40 (12.5)	20/110 (18.2)	0.591
*Astenia*	19/40 (47.5)	61/110 (55.4)	0.399
*Amnesia/neurological symptoms*	4/40 (10)	13/110 (11.8)	0.763
*Time to negative swab, days*	16.0 [8.0–25.0]	23.0 [15.0–30]	**0.006**

**Abbreviations:** CAD: coronary artery disease; CKD: chronic kidney disease; COPD: chronic obstructive pulmonary disease; KLS: ketoprofen lysine salt; and SD: standard deviation.

**Table 2 pharmacy-13-00065-t002:** Odds ratio of non-severe COVID-19 patients treated with ketoprofen lysine salt (KLS) group compared to cortico-steroids of developing main clinical outcomes: hospitalization, need for oxygen supplementation, and the presence of long COVID symptoms. The table included the odds ratio, the 95% confidence interval (IC), and the *p*-value for each clinical outcome.

	Odds Ratio	95% IC Value	*p*-Value
Hospitalization risk	0.140	0.0587–0.3421	**<0.001**
Need for oxygen supplementation	0.150	0.076–0.294	**<0.001**
Presence of long COVID symptoms	0.250	0.152–0.412	**<0.001**

**Table 3 pharmacy-13-00065-t003:** Multivariate analysis for (1) hospitalization risk, (2) oxygen support supplementation, and (3) persistence of post-COVID-19 symptoms after adjustments for significant variables. The table included the odds ratio, the 95% confidence interval (IC), and the *p*-value for each clinical outcome.

**Multivariate analysis for hospitalization risk**					
			**95% Confidence Interval**		
**Predictor**	**Standard Error**	**Odds-Ratio**	**Lower**	**Upper**	** *p* **
*Smoking Status (never versus current/former)*	0.432	0.4688	0.2011	1.093	0.079
*CAD or CHF* *(yes versus no)*	1453.530	1.67 × 10^−8^	0.0000	Inf	0.990
*Type II Diabetes* *(yes versus no)*	0.841	0.4126	0.0794	2.145	0.293
*Haematological Disorders* *(yes versus no)*	1771.930	3.63 × 10^−8^	0.0000	Inf	0.992
*Dyspnea/Chest tightness Yes versus No*	0.516	2.3344	0.8495	6.415	0.100
*Therapy Group: Ketoprofen Lysin Salt—Corticosteroid*	0.567	0.0796	0.0262	0.242	<0.001
**Multivariate analysis for oxygen support supplementation**					
			**95% Confidence Interval**		
**Predictor**	**Standard Error**	**Odds-Ratio**	**Lower**	**Upper**	** *p* **
*Smoking Status (never versus current/former)*	0.395	1.711	0.7883	3.713	0.174
*CAD or CHF* *(yes versus no)*	1383.976	5.23 × 10^−9^	0.0000	Inf	0.989
*Type II Diabetes* *(yes versus no)*	0.628	0.340	0.0994	1.166	0.086
*Haematological Disorders* *(yes versus no)*	1586.235	1.15 × 10^−8^	0.0000	Inf	0.991
*Dyspnea/Chest tightness Yes versus No*	0.584	15.155	4.8214	47.638	<0.001
*Therapy Group: Ketoprofen Lysin Salt—Corticosteroid*	0.384	0.127	0.0598	0.269	<0.001
**Multivariate analysis for post-COVID 19 symptoms**					
			**95% Confidence Interval**		
**Predictor**	**Standard Error**	**Odds-Ratio**	**Lower**	**Upper**	** *p* **
*Smoking Status (never versus current/former)*	0.290	0.847	0.480	1.495	0.567
*CAD or CHF* *(yes versus no)*	0.474	3.280	1.296	8.297	0.012
*Type II Diabetes* *(yes versus no)*	0.404	0.252	0.114	0.557	<0.001
*Haematological Disorders* *(yes versus no)*	0.566	3.182	1.049	9.655	0.041
*Dyspnea/Chest tightness Yes versus No*	0.318	1.810	0.970	3.377	0.062
*Therapy Group: Ketoprofen Lysin Salt—Corticosteroid*	0.280	0.324	0.187	0.562	<0.001

## Data Availability

The original contributions presented in this study are included in the article/Appendix A.

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
