# Peer review of "Ketoprofen Lysine Salt Versus Corticosteroids in Early Outpatient Management of Mild and Moderate COVID-19: A Retrospective Study"

_pharmacy, 2025, doi:10.3390/pharmacy13030065_

Round 1
Reviewer 1 Report
Comments and Suggestions for Authors
This retrospective observational study aims to assess the association between the administration of Ketoprofen Lysine Salt (KLS) as compared to corticosteroid and clinical outcomes in patients with mild-moderate COVID-19 afferent to an outpatient service.
In my opinion, the present analysis provides very weak evidence of a higher efficacy of KLS compared to corticosteroids. Moreover, the presentation of results is not accurate.
Please see the comments below:
- The main problem of course is confounding. The study is observational and thus a univariate (unadjusted) analysis of the association between treatment and outcomes is not sufficient to establish a causal relationship. A multivariable analysis (e.g. logistic regression) or propensity score based methods (matching or weighting) should be applied to try limiting bias due to baseline confounders. For instance, the higher proportion of patients with dyspnea/chest tightness at baseline in the corticosteroid group clearly influences the results on the main outcomes: i.e. the higher rates of hospitalizations and need for oxygen in the corticosteroid groups could be due (at least partially) to this worse baseline condition. Moreover, data about the time from occurrence of first symptom to the first visit could be included. The authors also acknowledge in the limitations that they could not retrieve information on other important prognostic factors as hematological parameters and markers of inflammation. At least, the analysis should be adjusted for measured potential confounders.
- More information about the treatments administered should be provided. What was the indication for prescribing KLS or corticosteroids? Did you measure the compliance to the treatment? Did any treatment switch happen and for which reasons? How about other concomitant treatments assumed by patients?
- When did the follow-up visit(s) take place? This is important especially for the outcome long COVID-19 symptoms. If the follow-up was not uniform for all patients, the incidence of persistency of symptoms could be wrongly estimated. If this is the case, a possibility is to analyze the outcome using incidence rates ratios or hazards ratios (measures that allow accounting for follow-up time) rather than relative risk / odds ratio.
- Presentation of results should be improved. Table 1 contains data about baseline characteristics and long COVID-19 symptoms but not on the other clinical outcomes. The caption of table 2 refers to relative risks but the table shows odds ratios (it would be better to use relative risks). Barplots of figure 1 (a,b,c) show absolute counts but it would be better to show percentages to compare the distribution between the two treatment groups. Moreover, in figure 1c the y axis is truncated, in figure 1d it should be explained what the vertical segment represent (standard deviation, standard error?). In the statistical analysis section, it is written that median and interquartile ranges were used to describe non-normally distributed variables but actually all continuous variables are described using means and standard deviations (e.g. I strongly suspect that duration of long COVID-19 symptoms does not have a symmetric distribution). The sentence “For non-normally distributed categorical variables, the Kruskal–Wallis test was used” is wrong because this test is suitable to compare non-normal continuous variables among three or more groups.
Author Response
This retrospective observational study aims to assess the association between the administration of Ketoprofen Lysine Salt (KLS) as compared to corticosteroid and clinical outcomes in patients with mild-moderate COVID-19 afferent to an outpatient service.
In my opinion, the present analysis provides very weak evidence of a higher efficacy of KLS compared to corticosteroids. Moreover, the presentation of results is not accurate.
Response: We wish to thanks the referee for the comments reported below. We have provided a significant revision of the manuscript – especially from the statistical standpoint – to manage all the comments/suggestions. We are very sorry for the previous inaccuracies and we hope you may appreciate the efforts in improving the quality of the manuscript and the strength of this research
Please see the comments below:
- The main problem of course is confounding. The study is observational and thus a univariate (unadjusted) analysis of the association between treatment and outcomes is not sufficient to establish a causal relationship. A multivariable analysis (e.g. logistic regression) or propensity score based methods (matching or weighting) should be applied to try limiting bias due to baseline confounders. For instance, the higher proportion of patients with dyspnea/chest tightness at baseline in the corticosteroid group clearly influences the results on the main outcomes: i.e. the higher rates of hospitalizations and need for oxygen in the corticosteroid groups could be due (at least partially) to this worse baseline condition.
- Response: We thank the referee for highlighting this remarkable criticism. We have prepared a major revision of the manuscript to tackle these flaws. In particular, we have now included multivariate analysis to assess the influence of each variable on the outcomes. As presented in the statistical analysis paragraph we included in the final multivariate analysis the factors with p value < 0.1 between the two groups. Final adjusted data support and strength our early analysis. Therefore, we appreciated this very important comment.
- Moreover, data about the time from occurrence of first symptom to the first visit could be included.
- Response: We wish to thank the referee for his/her remarkable comment. The study population included only patients who were referred to our clinic by 90 days after the negative swab. This important information was missing and we have now included it in the method paragraph. However, the absence of interval data between the negative swab and first visit to our clinic is an important study limitation and we have now discussed this important limitation.
- The authors also acknowledge in the limitations that they could not retrieve information on other important prognostic factors as hematological parameters and markers of inflammation. At least, the analysis should be adjusted for measured potential confounders.
- Response: We thank the referee this important comment. The study design – cross sectional observational non interventional – did not allow any speculation on serum biomarkers. This was out of the scope of this research despite we absolutely agree that would be very important. We can not exclude a possible difference in WBC, NLR, HB, PLT, CRP at baseline. Therefore, we have now included in the discussion paragraph this raised point as study limitation.
- More information about the treatments administered should be provided. What was the indication for prescribing KLS or corticosteroids? Did you measure the compliance to the treatment? Did any treatment switch happen and for which reasons? How about other concomitant treatments assumed by patients?
Response: We thank the reviewer for the comment. We have updated the Methods and Results sections to address this point. Additionally, the lack of a standardized assessment of treatment compliance represents a limitation of the study, which we have now included among the listed limitations.
- When did the follow-up visit(s) take place? This is important especially for the outcome long COVID-19 symptoms. If the follow-up was not uniform for all patients, the incidence of persistency of symptoms could be wrongly estimated. If this is the case, a possibility is to analyze the outcome using incidence rates ratios or hazards ratios (measures that allow accounting for follow-up time) rather than relative risk / odds ratio.
- Response: We wish to thank the referee for his/her remarkable comment. The study population included only patients who were referred to our clinic by 90 days after the negative swab. We decided to not include the interval between negative swab and the consultation visit as they might reflect other factors (availability of test, different interval for COVID-19 post clinic referral). This important information was missing and we have now included it in the method paragraph. For this reason we have decided to leave the RR.
- Presentation of results should be improved. Table 1 contains data about baseline characteristics and long COVID-19 symptoms but not on the other clinical outcomes. The caption of table 2 refers to relative risks but the table shows odds ratios (it would be better to use relative risks). Barplots of figure 1 (a,b,c) show absolute counts but it would be better to show percentages to compare the distribution between the two treatment groups. Moreover, in figure 1c the y axis is truncated, in figure 1d it should be explained what the vertical segment represent (standard deviation, standard error?). In the statistical analysis section, it is written that median and interquartile ranges were used to describe non-normally distributed variables but actually all continuous variables are described using means and standard deviations (e.g. I strongly suspect that duration of long COVID-19 symptoms does not have a symmetric distribution). The sentence “For non-normally distributed categorical variables, the Kruskal–Wallis test was used” is wrong because this test is suitable to compare non-normal continuous variables among three or more groups.
- Response: We wish to thank the referee for the raised issues and we are really sorry for the inaccuracies. In the revised version of the manuscript several changes to Tables and Figures have been made according your kind indications. In particular, we have now included in Table 1 the outcomes results. In the table 2 OR were substituted with RR. Bar-plots have been changed from absolute numbers to percentages. We have fixed figure 1c and better specified in the figure 1d that it represent the standard deviation. For the variables’ presentation, each variable has been assessed for normality distribution (see Kolmogorov Smirnov in the Supplementary Table) and presented as appropriate (mean+-SD or median IQR]). Finally, we are very sorry for the inaccuracy of the statistical analysis section – we have changed the Kruskal–Wallis which was wrongly written with the Spearman's rank correlation. The other outcomes were not put into Table 1 as it is very long actually and also because we have now included the multivariate analysis. Therefore, graphs reflect difference into other outcomes. However, we will happy to include them in Table 1 if necessary.
Reviewer 2 Report
Comments and Suggestions for Authors
Ketoprofen Lysine Salt vs Corticosteroids in early outpatients management in mild and moderate COVID-19- a retrospective study.
Good introduction and background to treatment of patients with mild COVID19 and the evolving nature of best practice
Methods and statistical analysis are relevant to study
Results clearly show 2 groups studied have similar baseline data. Results clearly show benefit of Ketoprofen Lysine Salt in reducing complications of Covid 19 compared to Corticosteroid use. Good discussion on NSAIS benefits in management of patients with Covid 19.
Discussion and conclusions drawn are appropriate from results.
Limitations of study as a retrospective study identified
Author Response
Ketoprofen Lysine Salt vs Corticosteroids in early outpatients management in mild and moderate COVID-19- a retrospective study.
Good introduction and background to treatment of patients with mild COVID19 and the evolving nature of best practice
Methods and statistical analysis are relevant to study
Results clearly show 2 groups studied have similar baseline data. Results clearly show benefit of Ketoprofen Lysine Salt in reducing complications of Covid 19 compared to Corticosteroid use. Good discussion on NSAIS benefits in management of patients with Covid 19.
Discussion and conclusions drawn are appropriate from results.
Limitations of study as a retrospective study identified
Response: many thanks for your comments.
Reviewer 3 Report
Comments and Suggestions for Authors
Some issues regarding the significance of the results are inherent to the study design, as for example election bias, as the allocation of treatment was not randomized. Also, the study population is drawn from a single-center cohort in Naples, Italy, raising concerns about external validity. Differences in healthcare settings, viral variants, and standard-of-care practices may limit the applicability of these findings to broader populations. A multi-center or international study design would enhance generalizability and credibility.
There are also some issues regarding confounding variables. The study does not adequately account for potential confounders such as variations in patient comorbidities, baseline inflammatory markers. Additionally, details on concurrent medications and potential interactions with KLS or corticosteroids are missing.
Statistical Limitations and Overinterpretation of Results - While p-values are reported, the article lacks confidence intervals for the key comparative outcomes, limiting the interpretation of the statistical significance. The authors should employ more robust statistical methods, including logistic regression models, to ensure a more comprehensive analysis.
Also, the article would benefit form an exploration of mechanistic pathways. The article provides a superficial discussion of the pharmacological mechanisms underlying KLS’s purported benefits over corticosteroids. While NSAIDs are known to modulate inflammatory pathways, their effects on viral dynamics and host immune response require deeper exploration. The article should incorporate in vitro or mechanistic studies to substantiate its claims regarding KLS’s potential.
Please consider rephrasing the whole paragraphs that have been used from external sources, and that have now caused a very high % for the plagiarism analysis.
In brief, the study raises awareness towards potential health benefits from using NSAID in the treatment in COVID, but the validity of the results is questionable.
Author Response
Some issues regarding the significance of the results are inherent to the study design, as for example election bias, as the allocation of treatment was not randomized. Also, the study population is drawn from a single-center cohort in Naples, Italy, raising concerns about external validity. Differences in healthcare settings, viral variants, and standard-of-care practices may limit the applicability of these findings to broader populations. A multi-center or international study design would enhance generalizability and credibility.
Response: We wish to thank the referee for this comment. We know the limits of our study due to non-randomized design and the retrospective nature of the statistical analyses. However, the results could provide the starting point for designing future well-designed large-scale clinical trials.
There are also some issues regarding confounding variables. The study does not adequately account for potential confounders such as variations in patient comorbidities, baseline inflammatory markers. Additionally, details on concurrent medications and potential interactions with KLS or corticosteroids are missing.
Response: We wish to thank the referee for this important comment. We have prepared a major revision of the manuscript to tackle these flaws. In particular, we have now included multivariate analysis to assess the influence of each variable on the outcomes. As presented in the statistical analysis paragraph we included in the final multivariate analysis the factors with p value < 0.1 between the two groups. Final adjusted data support and strength our early analysis. Also, the study design – cross sectional observational non interventional – did not allow any speculation on serum biomarkers. This was out of the scope of this research despite we absolutely agree that would be very important. Therefore, we have now included in the discussion paragraph this raised point as study limitation. The comorbidities we have now included in Table 1 their prevalence as in the previous version they were evaluated only as cumulative (Charlson Comorbidity Index)
Statistical Limitations and Overinterpretation of Results - While p-values are reported, the article lacks confidence intervals for the key comparative outcomes, limiting the interpretation of the statistical significance. The authors should employ more robust statistical methods, including logistic regression models, to ensure a more comprehensive analysis.
Response: We Thanks the referee for this valuable comment. In this version of the manuscript the statistical analysis has been substantially changed including IC and logistic regression. Furthermore, a final multivariate analysis has now been included to better understand the influence of variables. Finally, to avoid an overinterpretation of the results we have provided substantial changes. In the discussion paragraph and created a sub-paragraph about study limitations. Many thanks for these remarkable comments
Also, the article would benefit form an exploration of mechanistic pathways. The article provides a superficial discussion of the pharmacological mechanisms underlying KLS’s purported benefits over corticosteroids. While NSAIDs are known to modulate inflammatory pathways, their effects on viral dynamics and host immune response require deeper exploration. The article should incorporate in vitro or mechanistic studies to substantiate its claims regarding KLS’s potential.
Response: We thank the referee for this comment, we have added a detailed explanation about NSAIDs antiviral activity in the discussion (line 218): “Some NSAIDs have been reported, often at high doses, to have antiviral activity in vitro which has been supposed to contribute to their efficacy in the treatment of COVID-19 . For example, Xu et al. found that indomethacin has potent antiviral activity against SARS-CoV-2 in vitro and in vivo, and that recovery occurred significantly sooner with a combination of symptomatic treatment and indomethacin in CoV-infected dogs. Moreover, Sisakht et al. demonstrated that NSAIDs downregulated the expression of prostaglandin E synthase, and it’s well known that PGE2 increases the viral pathogenicity by contributing to hyperinflammatory as well as affecting the viral transcription, translation, and/or replication. In addition, the authors showed that NSAIDs acted as inhibitors of the 3CLpro, the main protease of the SARS- CoV-2 involved in the proteolytic maturation of the virus”.
Please consider rephrasing the whole paragraphs that have been used from external sources, and that have now caused a very high % for the plagiarism analysis.
In brief, the study raises awareness towards potential health benefits from using NSAID in the treatment in COVID, but the validity of the results is questionable.
Response: We thank the referee for this comment. Our data suggest that patients who received KLS during the early phase of infection had a significantly lower incidence of persistent post-COVID symptoms. However, we recognize some limitations in our study, such as the lack of medical history data that may have influenced the onset of COVID symptoms. Future studies are needed to establish a clearer causal relationship between different treatment for acute COVID-19 and the long-term consequences of the disease. Nevertheless, when comparing the corticosteroid and KLS treatment groups, a higher percentage of patients in the corticosteroid group required hospitalization and oxygen supplementation. Fewer patients in the KLS group showed persistent symptoms and for a shorter time. In conclusion, although our results suggest that KLS may be more effective than corticosteroids in reducing both the incidence and duration of COVID symptoms, further studies are necessary to validate these results and explore the factors that contribute to the long-term effects of COVID-19.
Round 2
Reviewer 1 Report
Comments and Suggestions for Authors
Despite the improvements implemented I still think that the issue of counfounding is too important to draw any reliable conclusion on the impact of KLS in reducing hospitalizations, need for oxygen supplementation and prolonged long-COVID-19 symptoms with respect to corticosteroids.
The statistical analysis is still weak and not accurate, some examples:
- In table 2 Odds ratios (KLS vs corticosteroids) are reported but still the term "Relative Risk" appears in the caption
- The multivariable logistic regression models reported in Table 3 are not reliable. The standard errors are extremely high for some coefficients (probably due to collinearity or separation issues). Moreover, the ORs for the treatment variable are reported as corticosteroids vs KLS in contrast to the choice made in table 2. It is also quite unnatural that exactly the same covairates were chosen for all the three outcomes.
- In figure 1 the joint percentages of treatment and outcome are reported. This is not very informative. The conditional percenteages of the outcome given the treatment should be reported (as those reported in the text in parargaph 4.2).
I suggest the authors to consult a professional statistician.
Author Response
Despite the improvements implemented I still think that the issue of counfounding is too important to draw any reliable conclusion on the impact of KLS in reducing hospitalizations, need for oxygen supplementation and prolonged long-COVID-19 symptoms with respect to corticosteroids.
The statistical analysis is still weak and not accurate, some examples:
- In table 2 Odds ratios (KLS vs corticosteroids) are reported but still the term "Relative Risk" appears in the caption
- Response: We wish to thank the referee for his/her valuable comment. We have change the wrong term with odds ratios as appropriate. We are very sorry for the inaccuracy.
- The multivariable logistic regression models reported in Table 3 are not reliable. The standard errors are extremely high for some coefficients (probably due to collinearity or separation issues). Moreover, the ORs for the treatment variable are reported as corticosteroids vs KLS in contrast to the choice made in table 2. It is also quite unnatural that exactly the same covairates were chosen for all the three outcomes.
- Response: We wish to thank the referee for his/her valuable comments. We have now included a supplementary table with collinearity analysis which document absence of significant collinearity. Alongside, we can't exclude separation issues which should be partially explained from the small number of patients (and subjects with positive outcome). This might influence the analysis and we have now included this very important consideration in the limitation paragraph. The choice of the covariates included in the multivariate analysis was made using a backward elimination procedure based on p value (see Methods). We have discuss these considerations into the study limitation paragraph.
- In figure 1 the joint percentages of treatment and outcome are reported. This is not very informative. The conditional percenteages of the outcome given the treatment should be reported (as those reported in the text in parargaph 4.2).
- Response:we have changed the figure 1 accordingly.
I suggest the authors to consult a professional statistician.
Reviewer 3 Report
Comments and Suggestions for Authors
I am satisfied with the revisions made by the authors. As such I feel the quality of the manuscript was improved to its potential, considering obvious limitations that have been mentioned.
Author Response
Many thanks for your comments.
Round 3
Reviewer 1 Report
Comments and Suggestions for Authors
The authors addressed the statistical issues raised and acknowledged the study limitations.
Still, that the evidence provided by this study that KLS is more efective than corticosteroids in reducing hospitalizations, need for oxygen supplementation and prolonged long-COVID-19 symptoms is very weak.